# A soil non-aqueous phase liquid (NAPL) flushing laboratory experiment based on measuring the dielectric properties of soil-organic mixtures via time domain reflectometry (TDR)

Alessandro Comegna[a*], Antonio Coppola[a], Giovanna Dragonetti[b], Angelo Sommella[c]

[a]School of Agricultural Forestry Food and Environmental Sciences (SAFE), University of Basilicata, Potenza, Italy.
[b]Mediterranean Agronomic Institute, Land and Water Division, IAMB, Bari, 70010, Italy.
[c]Department of Agriculture, University of Naples "Federico II", Italy.

*Correspondence to*: Alessandro Comegna (alessandro.comegna@unibas.it)

**Abstract.** The term non-aqueous phase liquid (NAPL) refers to a group of organic compounds with scarce solubility in water. They are the products of various human activities and may be accidentally introduced into the soil system. Given their toxicity level and high mobility, NAPLs constitute a serious geo-environmental problem. Contaminant distribution in the soil and groundwater entails fundamental information for the remediation of polluted soil sites. The present research explored the possible employment of time domain reflectometry (TDR) to estimate pollutant removal in a silt-loam soil that was primarily contaminated with a corn oil as a light NAPL and then flushed with different washing solutions. Known mixtures of soil and NAPL were prepared in the laboratory to achieve soil specimens with varying pollution levels. The prepared soil samples were repacked into plastic cylinders and then placed in testing cells. Washing solutions were then injected upward into the contaminated sample, and both the quantity of remediated NAPL and the bulk dielectric permittivity of the soil sample were determined. The above data were also used to calibrate and validate a dielectric model (the α mixing model) which permits the volumetric NAPL content ($\theta_{NAPL}$; m³/m³) within the contaminated sample to be determined and quantified during the different decontamination stages. Our results demonstrate that during a decontamination process, the TDR device is NAPL-sensitive: the dielectric permittivity of the medium increases as the NAPL volume decreases. Moreover, decontamination progression can be monitored using a simple (one-parameter) mixing model.

## 1. Introduction

Soil and groundwater contamination with NAPL from point or nonpoint sources is a severe problem of considerable complexity (Fitts, 2002; Fetter, 1993). The repercussions concern not only the deterioration of the soil's physical, mechanical and chemical properties, but also account for a potentially severe hazard to the well-being of humans and other living species (Freeze, 2000).

Soil flushing is the technical procedure used for treating polluted soils with water, surfactants and co-solvents (such as methanol, ethanol and propanols). Surfactant-enhanced flushing was developed from the conventional pump-and-treat

method. The success of this approach is related to the capacity of such chemical compounds to greatly enhance the
aqueous solubility of oils (Pennell et al., 1994; Parnian and Ayatollahi, 2008).
There is high interfacial tension between NAPL and water molecules that makes water a non-efficient cleaning material
in removing NAPL from the soil. Instead, surfactants and co-solvent agents can promote the enhanced removal of NAPL
from the subsurface through mobilization and solubilization (Martel et al., 1998; Rinaldi and Francisca, 2006; Parnian
and Ayatollahi, 2008).
Primary remediation entails the removal of the NAPL free phase by pumping. This extraction mechanism returns
appreciable effects if there is a region of high NAPL saturation. After primary pumping, a considerable portion of NAPL
remains constrained within the soil as capillary forces overcome viscous and buoyancy forces. This discontinuous NAPL
phase is referred to as trapped residual NAPL (or NAPL residual saturation), and its remediation is referred to as secondary
remediation (Parnian and Ayatollahi, 2008). Residual NAPL is a long-term source of soil and groundwater pollution
(Mercer and Cohen, 1990; Troung Hong and Bettahar, 2000).
To develop powerful decontamination procedures, the characterization of polluted soils is required. Practices usually
employed to characterize polluted soil sites are coring, soil sampling and the installation of monitoring wells for the
collection of water samples from aquifers (Mercer and Cohen, 1990). Since the above procedures are costly, different
dielectric techniques can be used to detect organic contaminants in soils. The most widely accepted geophysical technique,
based on the principle of electromagnetic wave (EMW) propagation, is ground penetrating radar (GPR; Knight, 2001).
Redman et al. (1991) described some field experiments in the application of GPR to detect NAPL plumes.
Rinaldi and Francisca (2006) used a coaxial impedance dielectric reflectometry (CIDR) technique to measure the complex
dielectric permittivity in sands contaminated by a paraffin oil. Their research into the dielectric behavior of NAPL-
contaminated soils during a decontamination process mainly focused on the removal efficiency of different washing
solutions, and on the spectral response of the contaminated medium during the various tests conducted.
TDR is a further geophysical device based on electromagnetic wave (EMW) principles that can also be used for this
purpose (Endres and Redman et al., 1993; Redman and De Ryck, 1994; Mohamed and Said, 2005; Moroizumi and Sasaki,
2006; Francisca and Montoro, 2012). Few experiments have been conducted coupling the TDR technique and NAPL. In
these studies estimation of NAPLs using TDR measurements of dielectric properties relies greatly on various mixing
models relating the measured dielectric permittivity to the volume fractions of the pore fluids and various soil phases such
as solid, water, air and NAPLs (van Dam et al., 2005).
Some interesting results were achieved by Persson and Berndtsson (2002) whilst investigating the influence of different
LNAPLs on TDR measurements in a homogeneous silica sand under saturated and unsaturated soil conditions.
Measurements of both dielectric permittivity and electrical conductivity allowed a method to be developed (the two-step
method) which measured the dielectric properties of the system against the amount of NAPL in soils. Comegna et al.
(2016) developed a general TDR-based methodology for evaluating the correlations between the dielectric response and
the NAPL content in variable saturated soils with different textures and pedological characteristics.
The purpose of this study was as follows: i) to investigate a possible extension of TDR technology to assess the effects
of NAPL removal in soils, and ii) revisit, on the basis of the acquired data and the experimental results, a dielectric model
to predict *"in real time"* the volumetric amounts of NAPL ($\theta_{NAPL}$) within the contaminated soil during the
decontamination process.
**2. Theoretical concepts of TDR**
TDR is a geophysical technique employed to determine the dielectric permittivity of liquids and solids (Ferrè and Topp,
2002, described this method in detail). In general, the bulk dielectric permittivity is a complex term ($\varepsilon_r^*$), which may be
expressed as follows (Robinson et al., 2003):

$$\varepsilon_r^* = \varepsilon'_r - j\left[\varepsilon''_r + \frac{\sigma}{\omega\varepsilon_0}\right] \tag{1}$$

where $\varepsilon'_r$ is the real part of dielectric permittivity, which gives the energy stored in the dielectrics at a certain frequency
and temperature, and $\varepsilon''_r$ is the imaginary part due to relaxations. The zero frequency conductivity $\sigma$, the angle frequency
$\omega$, the imaginary number $j = \sqrt{-1}$ and the permittivity $\varepsilon_0$ in free space contribute to define $\varepsilon_r^*$.
When the frequency of a TDR cable tester ranges between 200 MHz to 1.5 GHz, dielectric losses can be considered
minimal (Heimovaara, 1994) and the bulk dielectric permittivity $\varepsilon_b$ ($\cong$ the real part of permittivity) of a probe of length
$L$ is determined from the propagation velocity $v(=2L/t)$ of an electromagnetic wave along the wave guide across the
investigated medium by the following expression:

$$\varepsilon_b = \left(\frac{c}{v}\right)^2 \tag{2}$$

where $c$ (=$3 \times 10^8 \text{m s}^{-1}$) is the velocity of an electromagnetic wave in vacuum (Topp et al., 1980) and $t$ is travel time, i.e.
the time required by the generated signal to go back and forth through the TDR probe of length $L$ (m). This can be
calculated as follows:

$$t = \frac{2L}{c}\sqrt{\varepsilon_b} \tag{3}$$

The direct dependence of the signal's travel time $t$ upon soil dielectric permittivity is expressed by equation 3.
**3. Estimating volumetric NAPL content during a decontamination process in soils**
Dielectric mixing models, in their classical application, have been proposed to estimate the bulk dielectric permittivity of
a multi-phase medium, that is, a combination of three or four dielectric phases, and to couple the dielectric permittivity
of the medium to the dielectric permittivity of each single phase (Hilhorst, 1998). Recently, after analyzing the effects of
organic contaminants on soil dielectric properties, the above models were further developed to estimate the dielectric
properties of NAPL-polluted soils (Redman et al., 1991; Persson and Berndtsson, 2002; Francisca and Montoro, 2012,
Comegna et al., 2013a; Comegna et al., 2016; Comegna et al., 2017).
Based on such models, in the present study, we analyze the possibility of predicting the correlations between the
volumetric contents of NAPL ($\theta_{NAPL}$) and the dielectric response ($\varepsilon_b$) of contaminated soil during the progression of a
steady-state remediation process.
In the present research, we chose the so-called α model (Birchack et al., 1974; Knight and Endress, 1990; Roth et al.,

95    1990):

$$\varepsilon_b = \left[ \sum_{i=1}^{n} V_i \varepsilon_i^\alpha \right]^{1/\alpha} \tag{4}$$

where $V_i$ is the volume and $\varepsilon_i$ is the permittivity of each component of the complex medium; the exponent $\alpha$ is a fitting
parameter ($\alpha$ varies between -1 and 1), which may be related to the internal structure of the investigated medium (Hilhorst,
1998; Coppola et al., 2013; Coppola et al., 2015). Under the following hypothesis: i) the soil is homogeneous from a
textural point of view, and ii) the soil porosity ($\phi$) is constant, equation 4 was reformulated for our purposes.
For mixtures of soil ($s$) saturated with a certain amount of washing solution ($ws$), in rearranging the model formulation of
Rinaldi and Francisca (2006), the $\alpha$ model yields the following:

$$\varepsilon_{s-ws}^\alpha = [(1 - \phi)\varepsilon_s^\alpha + \phi\varepsilon_{ws}^\alpha] \tag{5}$$

where $\varepsilon_{s-ws}$ is the soil-washing solution permittivity, and $\varepsilon_s$ and $\varepsilon_{ws}$ are the permittivities of soil particles and washing
solutions, respectively. By the same token, for soil organic ($s$-NAPL) compounds at saturation, the $\alpha$ model can be
expressed as follows:

$$\varepsilon_{s-NAPL}^\alpha = [(1 - \phi)\varepsilon_s^\alpha + \phi\varepsilon_{NAPL}^\alpha] \tag{6}$$

where $\varepsilon_{s-NAPL}$ is the permittivity of the soil-NAPL mixture, and $\varepsilon_{NAPL}$ is the oil permittivity.
A medium consisting of soil particles, washing solution and NAPL ($s$-$ws$-NAPL) can be viewed as a mix of soil-washing
solution (equation 5) and soil-NAPL (equation 6):

$$\varepsilon_{s-ws-NAPL}^\alpha = [\beta\varepsilon_{s-NAPL}^\alpha + (1 - \beta)\varepsilon_{s-ws}^\alpha] \tag{7}$$

where $\beta$ is the relative volume of NAPL contained in the whole fluid phase:

$$\beta = \frac{\theta_{NAPL}}{(\theta_{ws} + \theta_{NAPL})} = \frac{\theta_{NAPL}}{\theta_f} \tag{8}$$

where $\theta_f$ is the volumetric fluid content (m³/m³), sum of the volumetric washing solution content ($\theta_{ws}$) and volumetric
NAPL content ($\theta_{NAPL}$); $\beta$ varies between 0 (i.e. a soil-washing solution mixture) and 1 (i.e. a soil-NAPL mixture).
To estimate $\theta_{NAPL}$, equation 7 is first reformulated in terms of $\beta$:

$$\beta = \frac{\varepsilon_{s-ws}^{\alpha} - \varepsilon_{s-ws-NAPL}^{\alpha}}{\varepsilon_{s-ws}^{\alpha} - \varepsilon_{s-NAPL}^{\alpha}} = \frac{(1-\phi)\varepsilon_s^{\alpha} + \phi\varepsilon_{ws}^{\alpha} - \varepsilon_{s-ws-NAPL}^{\alpha}}{\left((1-\phi)\varepsilon_s^{\alpha} + \phi\varepsilon_{ws}^{\alpha}\right) - \left((1-\phi)\varepsilon_s^{\alpha} + \phi\varepsilon_{NAPL}^{\alpha}\right)} \tag{9}$$

Substituting equation 8 into equation 9, and considering that for a saturated medium, the volumetric fluid content is equal
to soil porosity (i.e. $\theta_f = \phi$), $\theta_{NAPL}$ can be calculated as follows:

$$\theta_{NAPL} = \frac{(1-\phi)\varepsilon_s^{\alpha} + \phi\varepsilon_{ws}^{\alpha} - \varepsilon_{s-ws-NAPL}^{\alpha}}{\varepsilon_{ws}^{\alpha} - \varepsilon_{NAPL}^{\alpha}} \tag{10}$$

Equation 10 correlates the dependence of volumetric NAPL content with soil porosity; $\theta_{NAPL}$ can be estimated (within
the contaminated soil) during the progression of a remediation process once the dielectric permittivity of the soil-
contaminated mixture ($\varepsilon_{s-ws-NAPL}$) is known.
**4 Materials and Methods**
**4.1 Soil and fluid properties**
A silt-loam Anthrosol (IUSS Working Group WRB, 2006) from the region of Puglia (Italy) was used for this study. The
soil texture was measured by means of the hydrometer method (Day, 1965), while the Walkley–Black procedure (Allison,
1965) was used to determine soil organic $C$ content. The method developed by Miller and Curtis (2006) was used to
measure soil electrical conductivity ($EC_w$), while soil $pH$ was determined on the basis of a 1:1 soil/water ratio (Eckert,
1988). In textural terms, the soil comprised 15.7% sand, 11.6% clay and 72.4% silt. Soil porosity was 0.57%, organic
content 1.84%, $EC_w$ 0.17 dS/m and soil $pH$ 8.40.
The NAPL employed for the laboratory tests was corn oil ($\varepsilon_{NAPL}$=3.2; $EC_{NAPL}$=0.055 dS/m at 25°C) with a density of 0.905
g/cm³ (at 25°C). Three different removal solutions were employed for soil cleaning: a) a first solution (referred to below
as wd) composed of 99% distilled water and 1% commercial detergent ($\varepsilon_d$=9.22, at 25°C), b) a second solution (wda#1)
composed of 90% distilled water, 1% commercial detergent and 9% methanol as co-solvent ($\varepsilon_{alcohol}$=26.13, at 25°C) and
c) a third solution (wda#2) composed of distilled water (85%) with commercial detergent (1%) and methanol (14%). The
dielectric permittivity of the washing solutions, measured at 25°C, was $\varepsilon_{wd}$=75.04, $\varepsilon_{wda\#1}$=68.98 and $\varepsilon_{wda\#2}$=65.92, whereas
the dielectric permittivity of the tested soil saturated with each of the three cleaning solutions was $\varepsilon_{soil+wd}$=34.59,
$\varepsilon_{soil+wda\#1}$=31.04 and $\varepsilon_{soil+wda\#2}$=30.10.
**4.2 Measurement of dielectric permittivity of soil-NAPL contaminated samples during soil remediation**
**4.2.1 Experimental setup**
As illustrated in Fig. 1, the experimental layout consisted of the following: i) a Techtronix (model 1502C) cable tester; ii)
a three-rod probe 14.5 cm long with a wire diameter of 0.003 m and a wire spacing of 0.02 m, introduced vertically into
the soil samples; iii) a testing cell 0.15 m high and 0.08 m in diameter; iv) a peristaltic pump used for upward movement
of the washing solution.
**4.2.2 Sample preparation and testing procedures**
Soil was oven dried at 105°C and passed through a 2-mm sieve. Known amounts of soil and oil were mixed together,
shaken and then kept for 24 hours in sealed plastic bags to avoid any evaporation and ensure a uniform distribution of oil
within the sample and good oil adsorption by the soil matrix. The samples were then allocated to cylindrical boxes. With
a view to achieving different degrees of (initial) soil contamination, volumetric NAPL content ($\theta_{NAPL}$) was varied from
0.05 to 0.40 (in steps of 0.05). In all, each washing solution comprised eight oil-contaminated soil samples.
For all experiments, the soil samples were placed in the vessels in various steps at a bulk density of 1.13 g/cm³. During
TDR measurements, the soil samples were conserved at a temperature of 25°C by using a thermostat box. Remediation
was performed using an upward flux of diverse pore volumes $T$ of three washing solutions (wd, wda#1 and wda#2)
supplied at the rate of 90 cm³/h, corresponding to a Darcian velocity of 1.8 cm/h. After collection of the outflow from the
soil columns, the surnatant NAPL was separated from the washing solution and the quantity of NAPL remediated from
the soil was determined.
The obtained data series were employed to calibrate the proposed dielectric model of equation 10. An independent data
set, obtained in the same manner as the calibration data set, was used for model validation.
**4.3. Numerical indices for model performance evaluation**
The goodness of equation 10 was evaluated using two different criteria: i) the mean bias error (*MBE*), and ii) the model
efficiency (*EF*), computed according to the following relations (Legates and McCabe Jr, 1999):

$$MBE = \frac{\sum_{i=1}^{N}(E_i - O_i)}{N} \tag{11}$$


$$EF = 1 - \frac{\sum_{i=1}^{N}(E_i - O_i)^2}{\sum_{i=1}^{N}(O_i - \overline{O})^2} \tag{12}$$

where $E_i$ and $O_i$ are respectively the expected and the observed value, $\overline{O}$ is the mean of the observed data, and $N$ is the
number of observations.
*MBE* measures the differences between model-simulated data and measured values (positive *MBE* values are used to
indicate average overprediction, while negative values indicate underprediction). The model's ability to forecast $\theta_{NAPL}$ is
described by parameter *EF*, according to which *EF*=1 indicates perfect accord between predicted and measured data.
**5. Results and Discussion**
**5.1 Influence of washing solution on NAPL removal**
Figures 2a, b, c, d, e and f, with reference to the most representative experimental results, reveal the influence of pore
volumes $T$ on evaluated bulk dielectric permittivity ($\varepsilon_{s-ws-NAPL}$) for the soil specimens initially polluted with oil. As the
washing solution started to remove oil, the dielectric permittivity rose due to the larger dielectric permittivity of the
flushing mixture. As the remediation solution continued to move upward, the rising rate of the dielectric permittivity
decreased and asymptotically approached a constant value. This steady value was smaller than that observed when the
soil specimens were completely saturated by only the flushing solution (i.e. wd, wda#1 or wda#2), which in our tests
corresponds to the condition of a completely decontaminated soil. This difference in values is undoubtedly due to oil
confined in soil pores (i.e. NAPL residual saturation). For the same reason, residual saturation may explain why
insignificant oil remediation was observed for $\theta_{NAPL}$ values less than 0.15. This aspect may be explained by the fact that
for low volumetric NAPL contents, the non-wetting fluid (oil) is disconnectedly distributed (i.e. immobile) in the soil
samples, which means that $\theta_{NAPL}$ is close to the limiting *residual value*, and thus NAPL loses its ability to move in the
soil in response to a hydraulic gradient [i.e. capillary retention forces are greater than gravitational forces, which tend to
immobilize the NAPL (Brost and DeVaull, 2000)].
The NAPL volumes removed for different washing solutions and the initial volumetric content of NAPL are compared in
Fig. 3. For all the three cleaning solutions adopted, the experiments ultimately demonstrate (for a fixed $\theta_{NAPL}$) the same
results in terms of soil decontamination, and they show that NAPL removal increases with increasing $\theta_{NAPL}$. In some cases
(i.e. $\theta_{NAPL}$=0.15, 0.20 and 0.30), contaminated samples flushed with the wda#1 solution yield slightly higher removal
efficiency values compared to the samples flushed with wd and wda#2. Martel et al. (1998) suggest the need to investigate
the best water-surfactant-alcohol combination in order to enhance NAPL solubilization in soil.

**5.2 Model calibration and validation**

For the model (equation 10) calibration methodology, with reference to the three washing solutions (wd, wda#1 and
wda#2), we analyze the effect of the measured dielectric permittivity on volumetric NAPL content ($\theta_{NAPL}$) in order to
estimate the $\alpha$ parameter of the model. The complete calibration data set of estimated $\alpha$ parameters is reported in Table
1. The $\alpha$ parameter of the mixing model was determined, for a fixed $\theta_{NAPL}$ value and washing solution, by an optimization
procedure based on the least squares technique, and was kept constant for each of the remediation tests developed.
A permittivity value of 3.70 was adopted for the solid phase. This value was determined using the *"immersion method"*
which is commonly employed for estimating the $\varepsilon_s$ of soils (Robinson and Friedman, 2003; Kameyama and Miyamoto,
2008; Comegna et al., 2013b).
For the sake of brevity, a selection of the experimental $\varepsilon_{s\text{-}ws\text{-}NAPL}$-$\theta_{NAPL}$ relationships (validation dataset) is reported in Fig.
4a, b, c, d, e and f. The data in Fig. 4 (except for Fig. 4e, f) show that some of the model-simulated values tend to
overestimate the measured data. This behavior is mostly restricted to the beginning of the remediation process, when a
rapid change in dielectric permittivity may be observed. This behavior was also verified in other tests (not shown here)
and may be explained by invoking both NAPL properties such as liquid density, surface tension and viscosity, and soil
properties including moisture content, relative permeability and soil porosity (Brost and DeVaull, 2000; Wang et al.,

198    2013).

Mercer and Cohen (1990) referred to the existence, in NAPL-contaminated soils, of a *"double fluid domain,"* defined as
the composition of the following: i) mobile pools, which are NAPL-connected phases that move in the soil and ii)
immobile residuals (i.e. low permeability regions), which depend on small disconnected blobs or ganglia within the
contaminated soil (see also section 5.1 above). As long as the flushing continues, mobile pools are reduced and the oil
tends increasingly to be trapped in the immobile areas. This means that, during soil cleaning, the capacity of non-wetting
fluids to respond to gravitational forces gradually diminishes (Luckner et al., 1989). From a dielectric point of view, this
mechanism may appear as a rapid dielectric permittivity increase (identified in Fig. 4 as *fast oil mobility region*) within a
few pore volumes. When this fast mobility mechanism is dominant, the predictions of equation 10 fail.
Another possible explanation for this discrepancy between the observed and the predicted permittivity values may be
linked to the propensity of NAPL-water mixtures to form macroinclusions in the soil (Persson and Berndtsson, 2002),
which affected the initial pore-scale distribution of NAPL, and thus the global dielectric response of the medium (Ferré
et al, 1996), during the first remediation stages.
However, since the phenomenon is mostly limited to the initial part of the washing process, overall model effectiveness
is not compromised, as also shown in Table 2, which summarizes the goodness-of-fit statistical indices, and in Fig. 5a, b,
c, d, e, f, where the estimated $\theta_{NAPL}$ from equation (10) and the known $\theta_{NAPL}$ are illustrated in a series of 1:1 scatter plots.
Overall, both graphical and quantitative evaluations in terms of *MBE* and *EF* reveal the suitability of the dielectric model
adopted to estimate the volumetric NAPL content in the $\theta_{NAPL}$ range 0.15-0.40.
**6. Conclusions**
This paper presented an extensive dataset of remediation experiments that were conducted at a laboratory scale using corn
oil as a soil contaminant, and three different solutions for soil cleaning. The results of these tests were employed to
investigate the potential of the TDR technique in monitoring the development of a steady-state decontamination process.
Dielectric data analysis showed that, during soil flushing, dielectric permittivity behavior is highly dependent on the initial
volumetric content and intrinsic permittivity of the specific NAPL: *removal of NAPL produces an increase in bulk*
*dielectric permittivity*, due to the low value of oil permittivity. The experiments conducted also allowed us to calibrate
and validate a dielectric mixing model (equation 10). The model outcomes are encouraging; the calculated statistical
indices confirmed a high accuracy in NAPL predictions of the $\alpha$-model at different stages during soil cleaning, with the
only exception of the very initial cleaning stage (confined to the low values of *T*) where the eventual presence of a *fast*
*flow region* may limit its applicability.
The approach requires additional experiments and data sets for model calibration and validation in different pedological
contexts, mainly to confirm the potential of the methodology developed. Furthermore, an effort should be made,
introducing the water phase, *ab initio* in the experimental setup, to simulate a possible natural contamination-
decontamination scenario more accurately. Finally, full field-scale tests should also be conducted to evaluate the
performance of equation 10 in real field conditions.
*Data availability*. The dataset used in this paper is available on request to alessandro.comegna@unibas.it.
*Competing interests*: The authors declare that they have no conflict of interest.

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

**Figures**

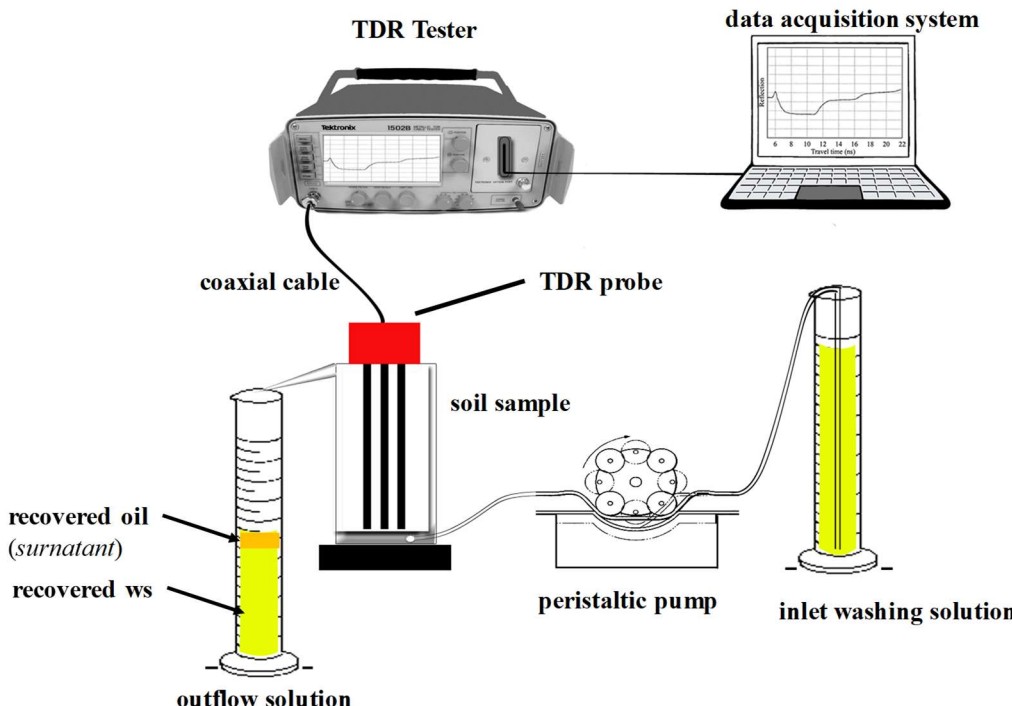


**Figure 1. Experimental setup used in the NAPL removal experiments (from Comegna et al., 2013c).**

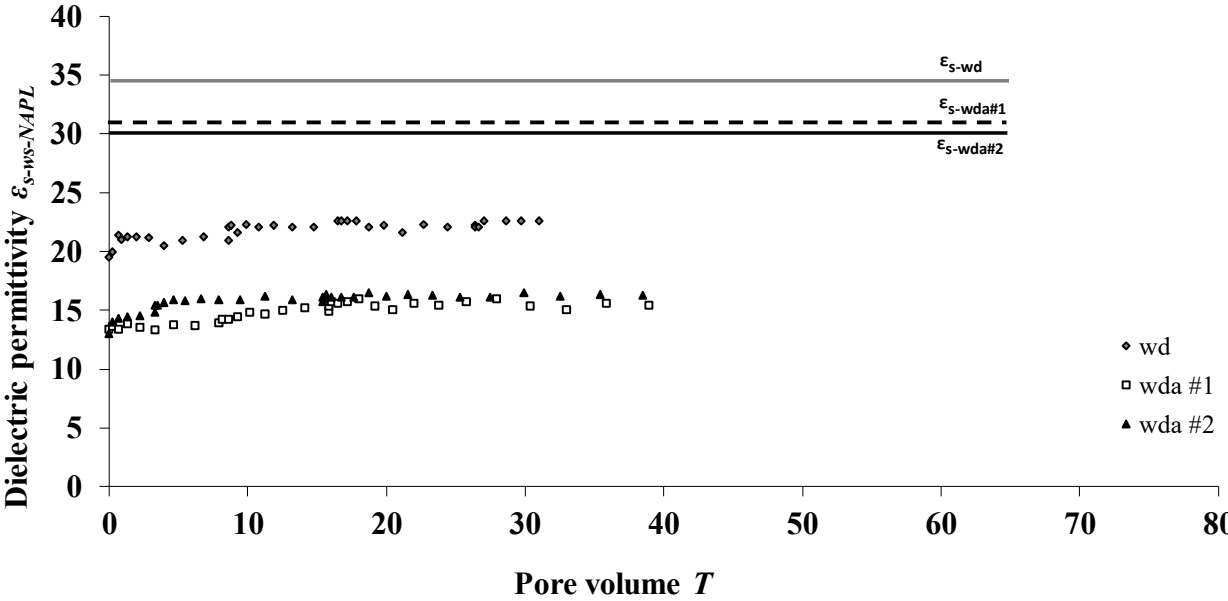

a) $\theta_{NAPL}$=0.15


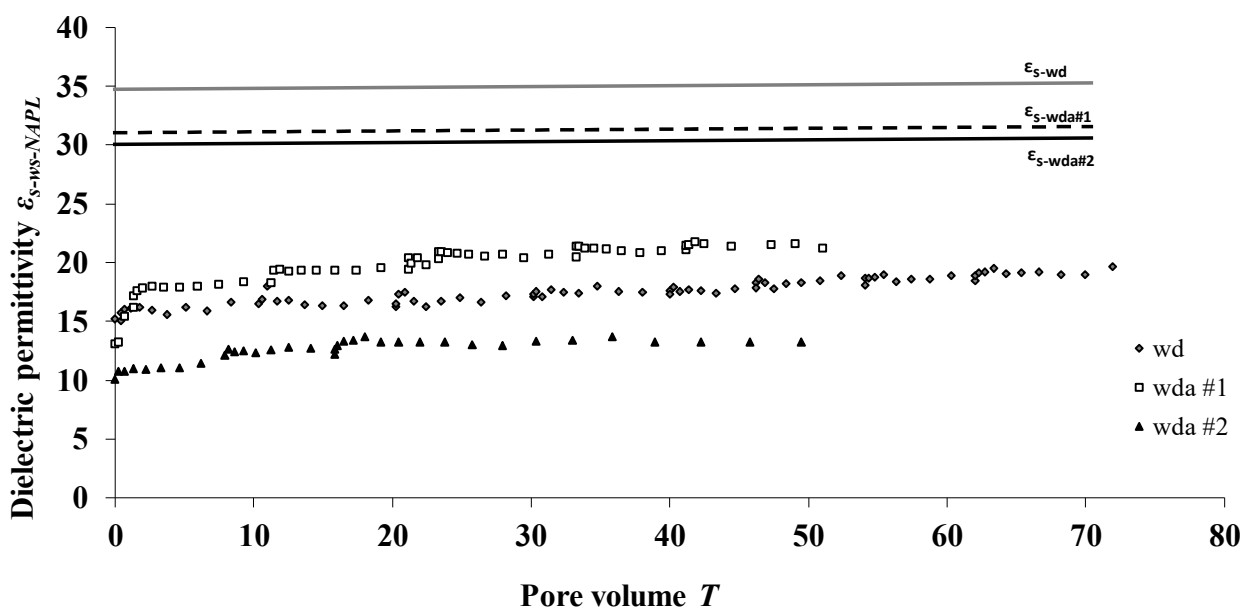

b) $\theta_{NAPL}$=0.20


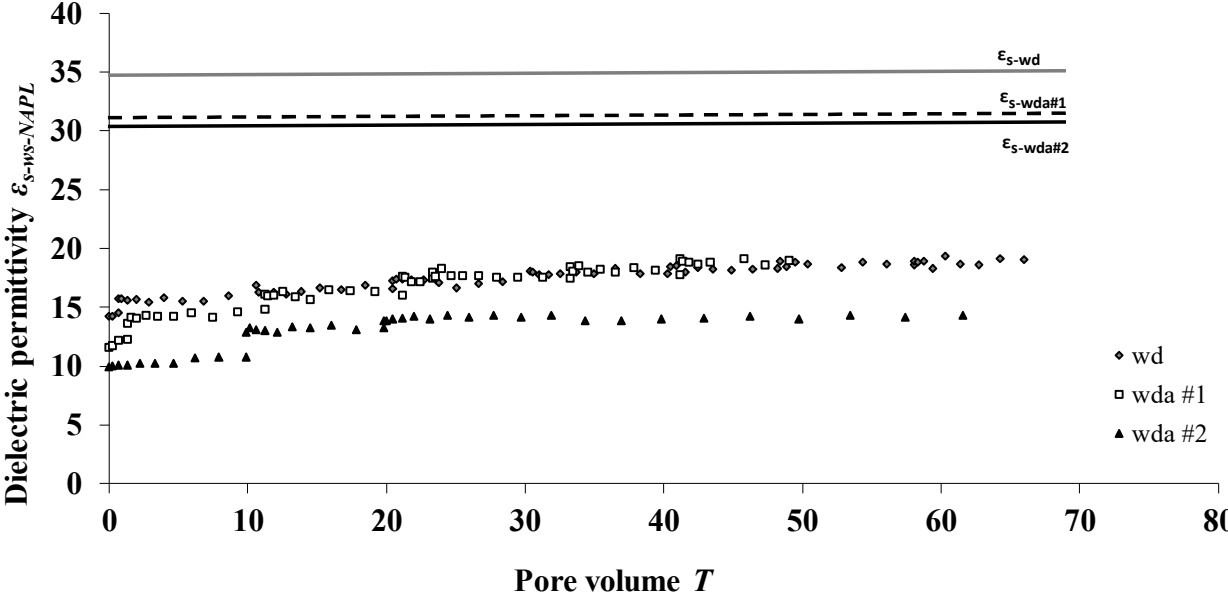

c) $\theta_{NAPL}=0.25$

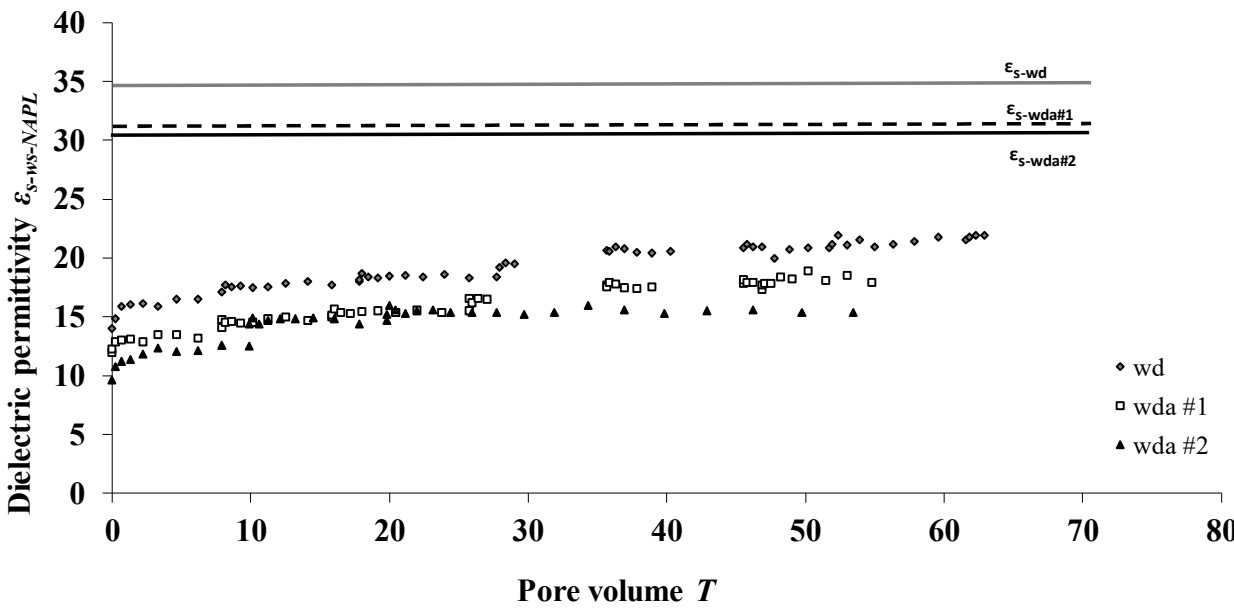

d) $\theta_{NAPL}=0.30$



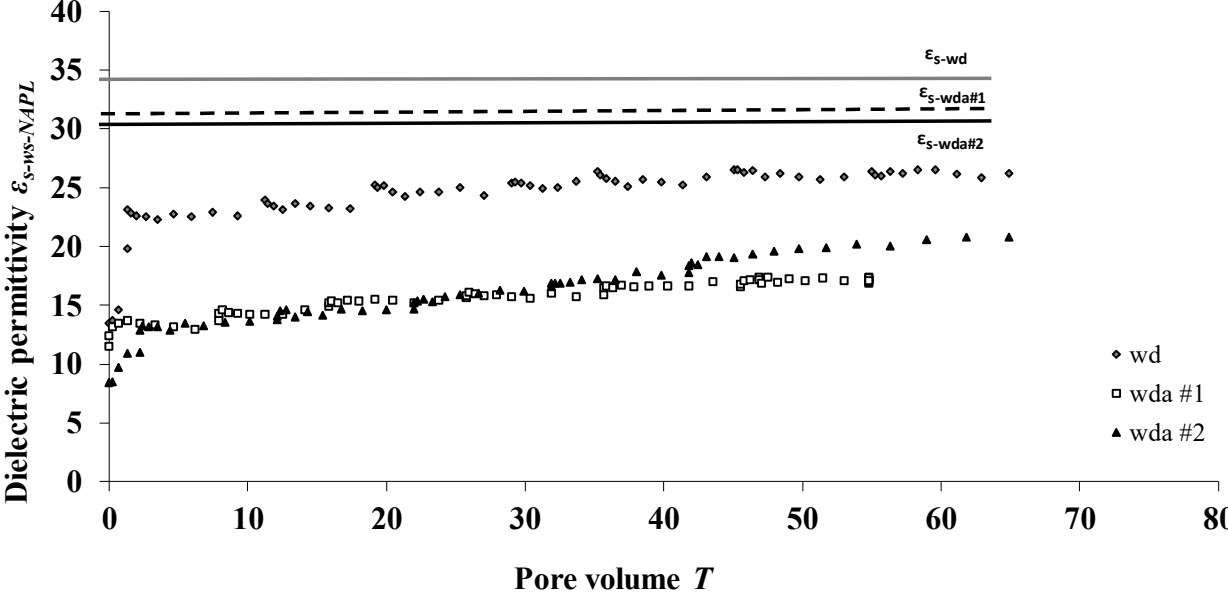

e) $\theta_{NAPL}=0.35$

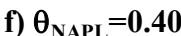

f) $\theta_{NAPL}=0.40$

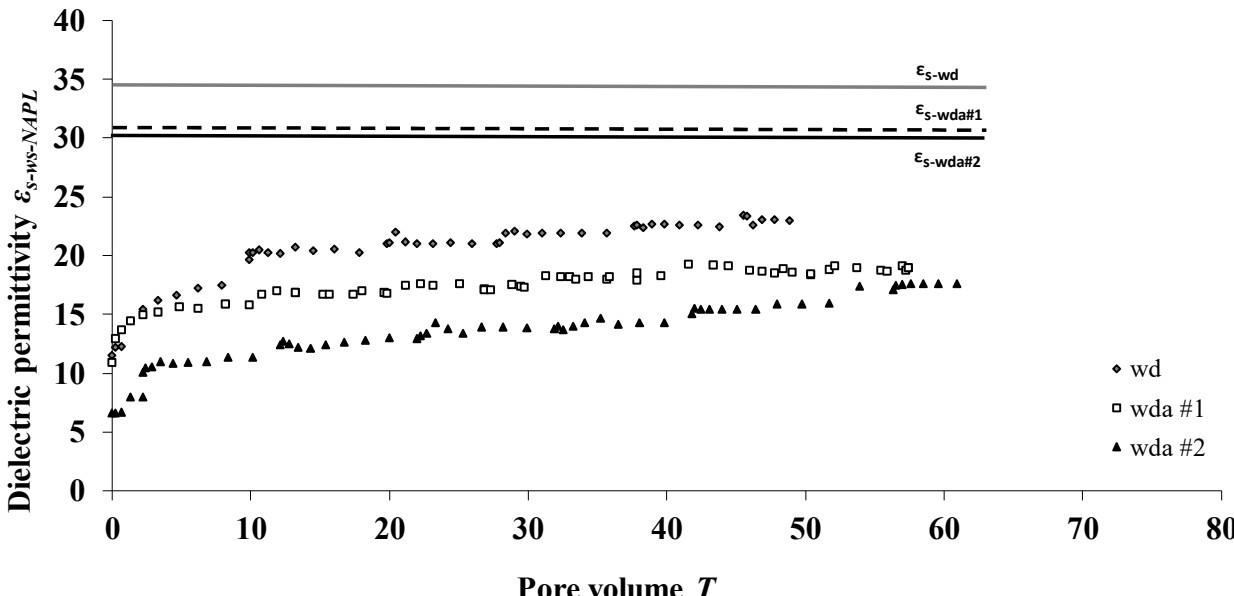


**Figure 2. Selection of experimental relationships between the measured dielectric permittivity ($\varepsilon_{s-ws-NAPL}$) and number of pore**
**volumes $T$ under the effect of different washing solutions: i) water-detergent (wd) and ii) water-detergent-alcohol (wda#1 and**
**wda#2).**

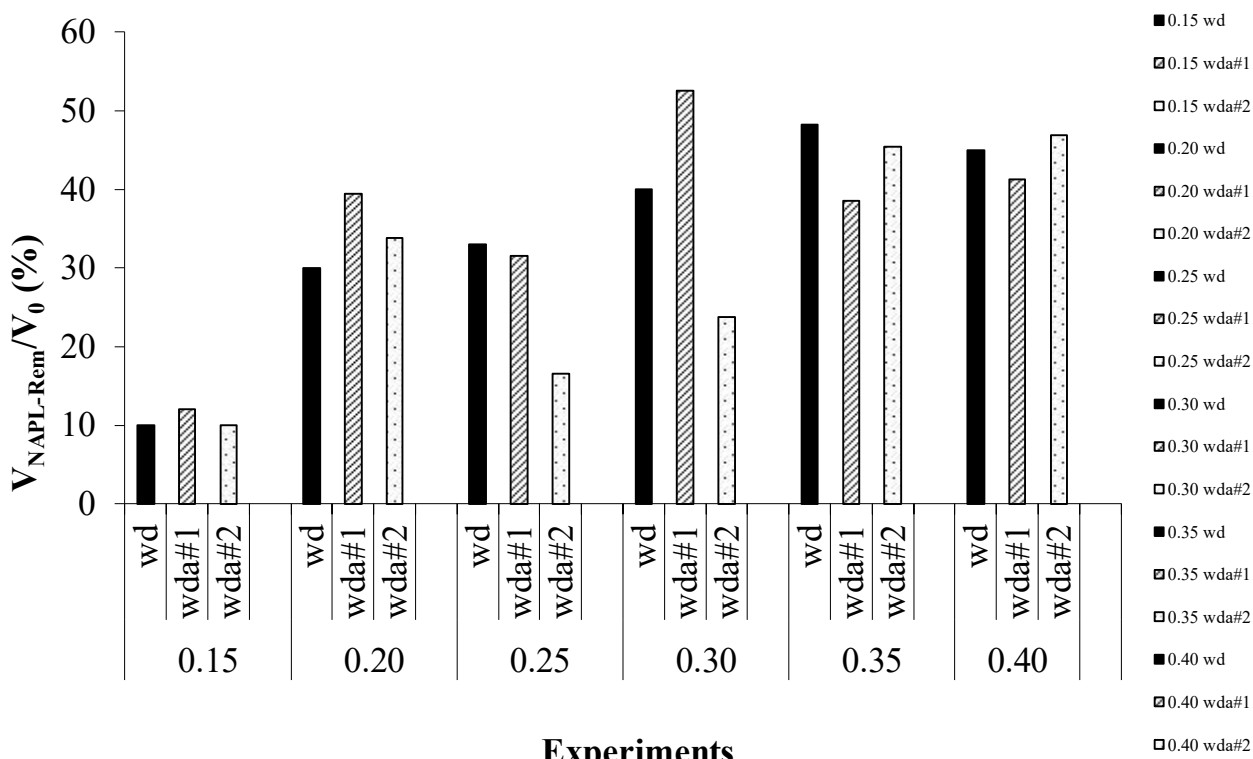


**Figure 3.** Volume of NAPL recovered ($V_{NAPL-Rem}$) with respect to the initial volume of NAPL present in the soil sample ($V_0$) of

different washing solutions (wd, wda#1 and wda#2) for different experiments ($\theta_{NAPL}$=0.15, 0.20, 0.25, 0.30, 0.35, 0.40).

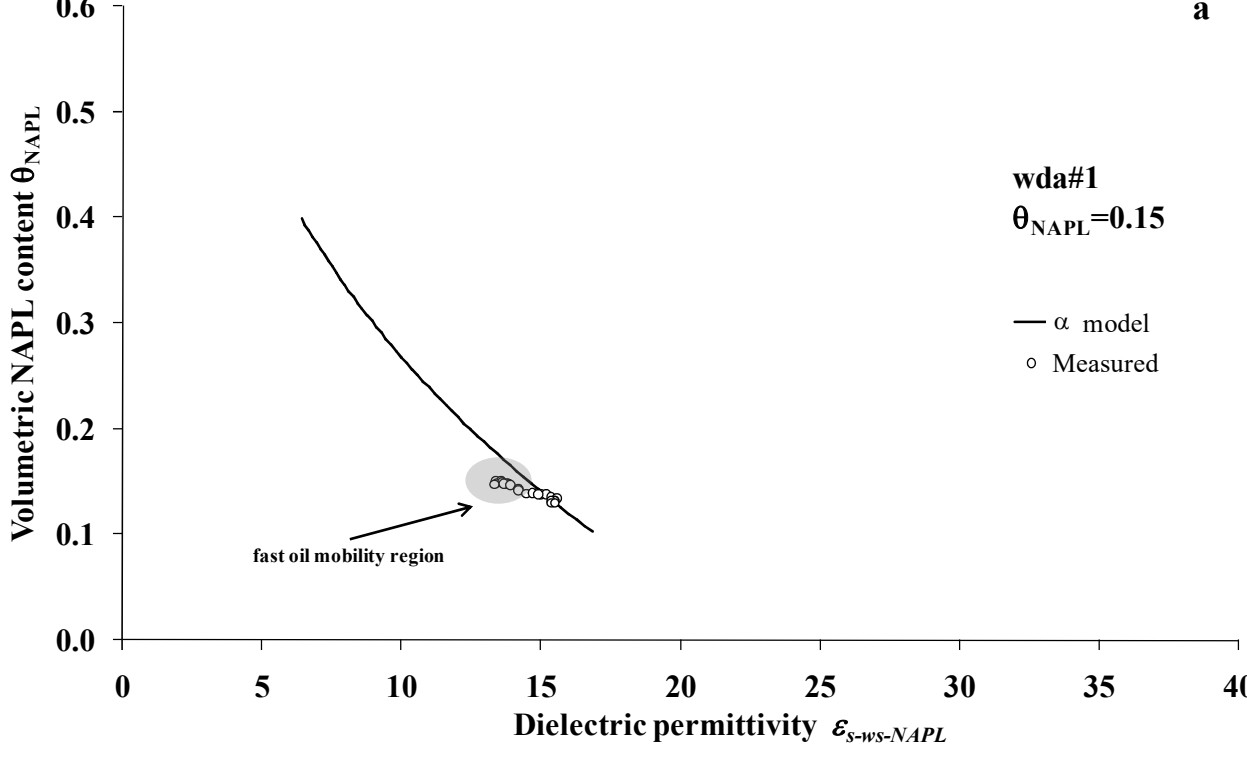


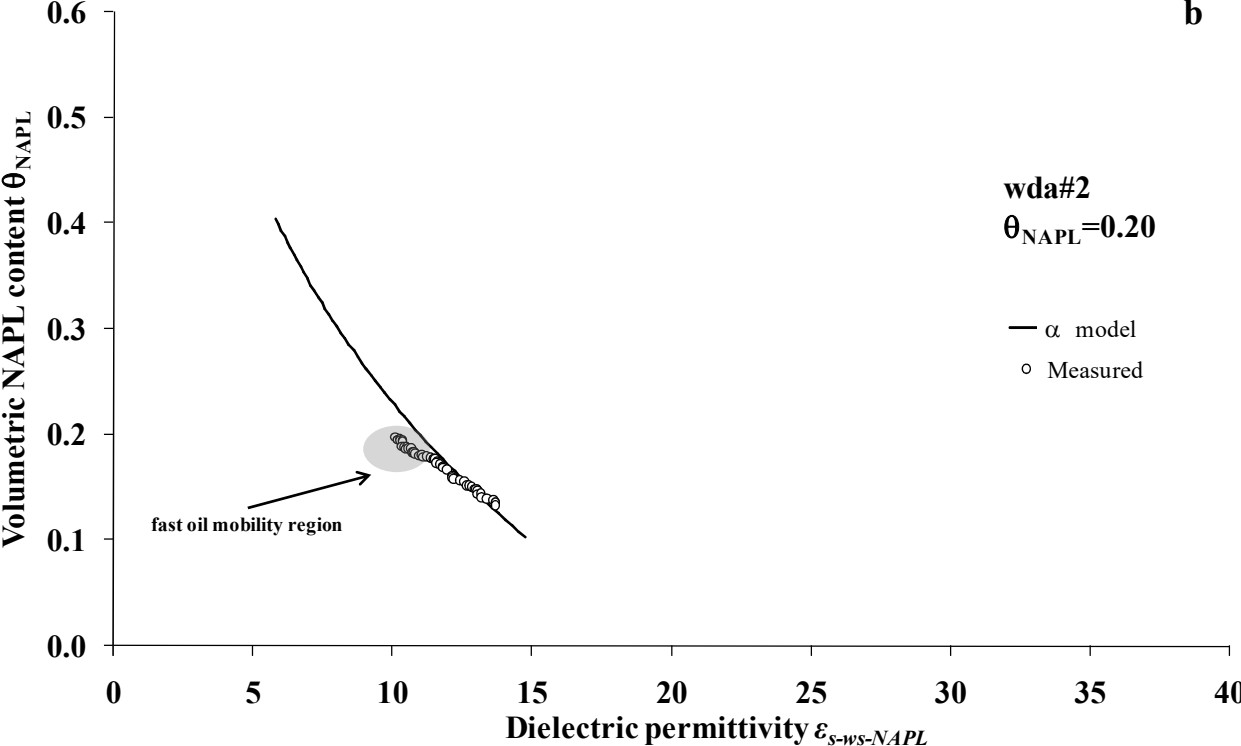


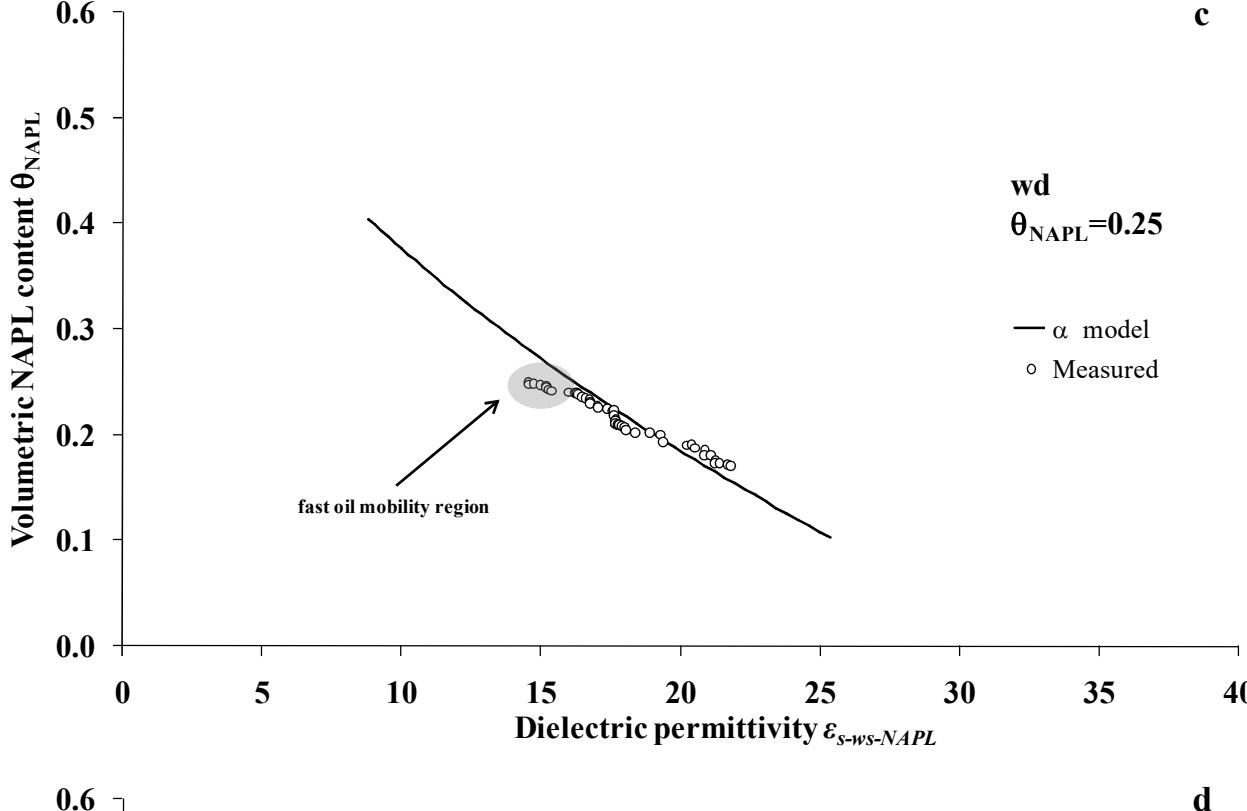

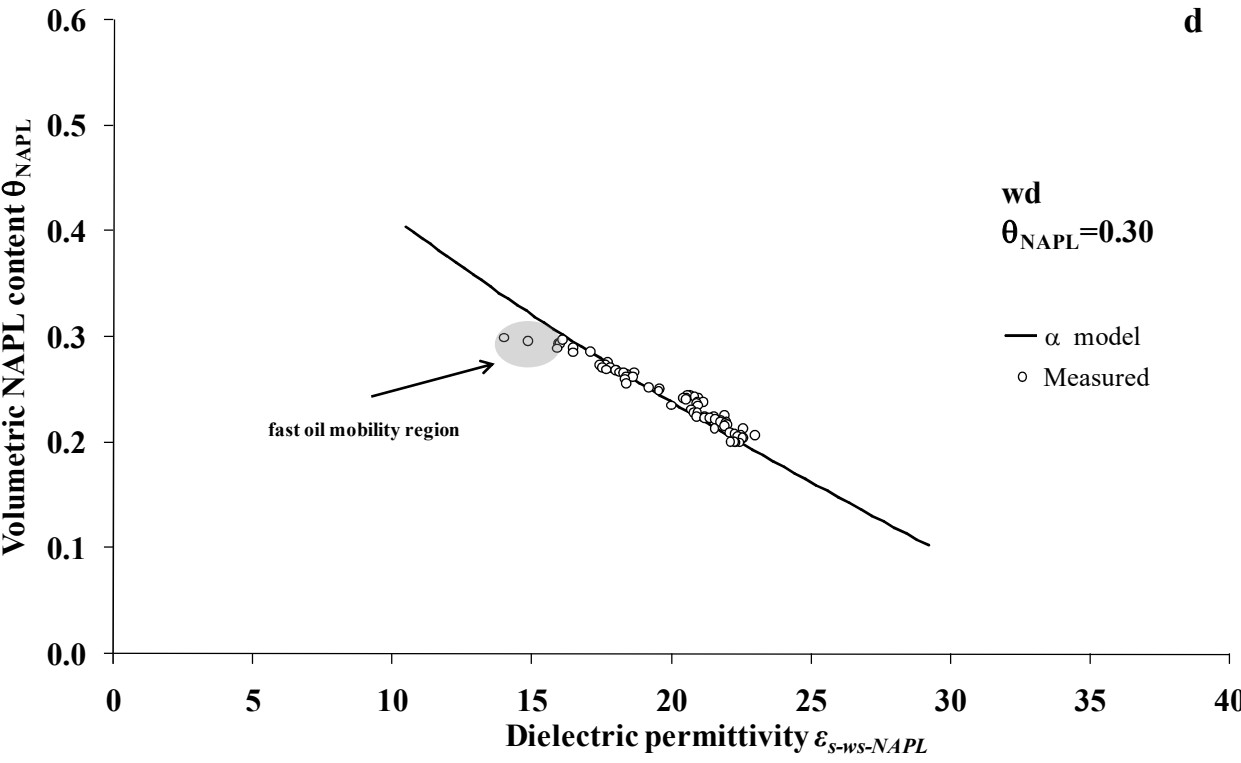

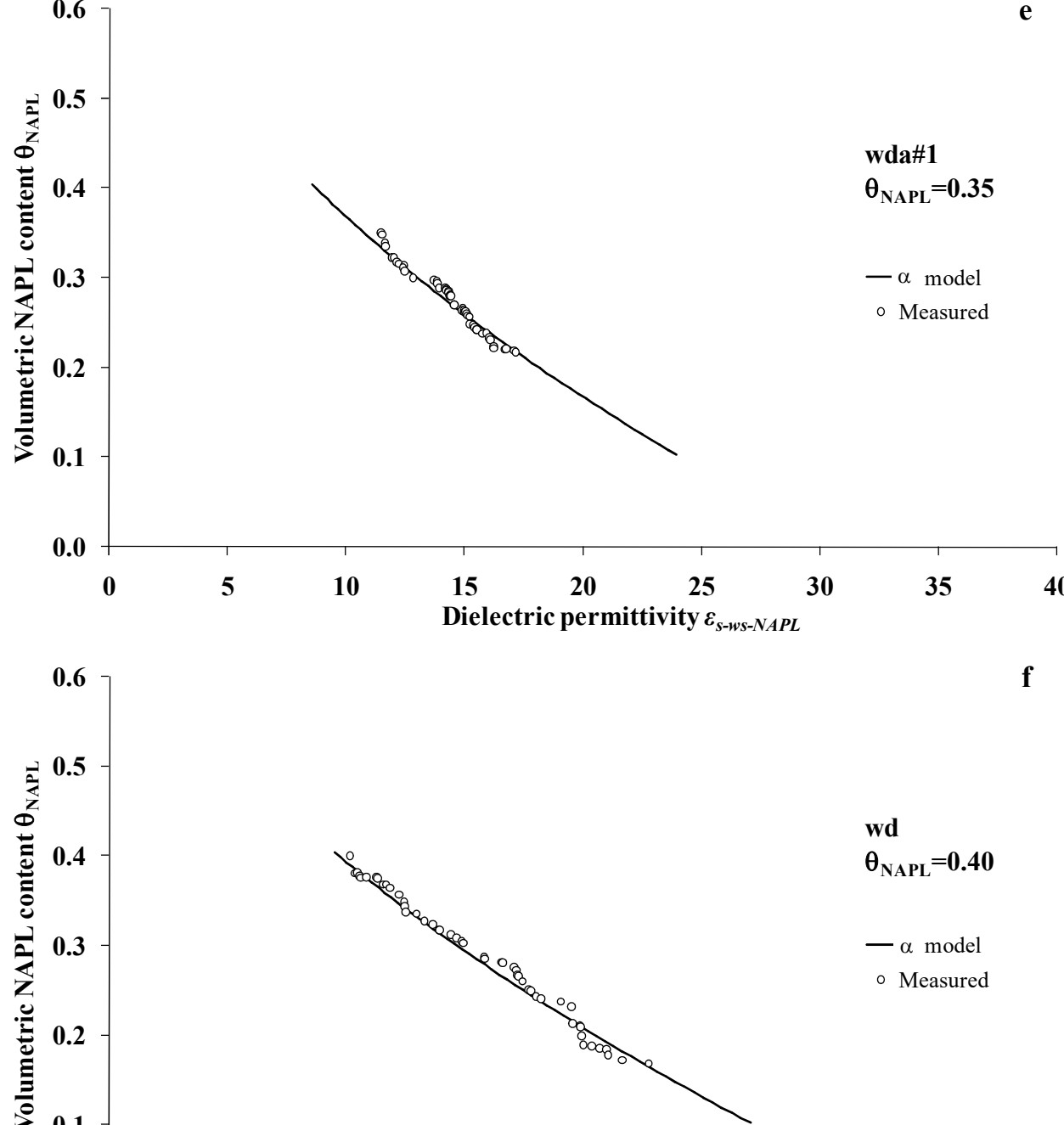


**Figure 4 a, b, c, d, e, f. Selection of observed (symbols) and modeled (dashed lines) volumetric NAPL content (θ$_{NAPL}$) versus**
**dielectric permittivity (ε$_{s-ws-NAPL}$), with reference to the three washing solutions (wd, wda#1 and wda#2) used during the**
**remediation tests.**

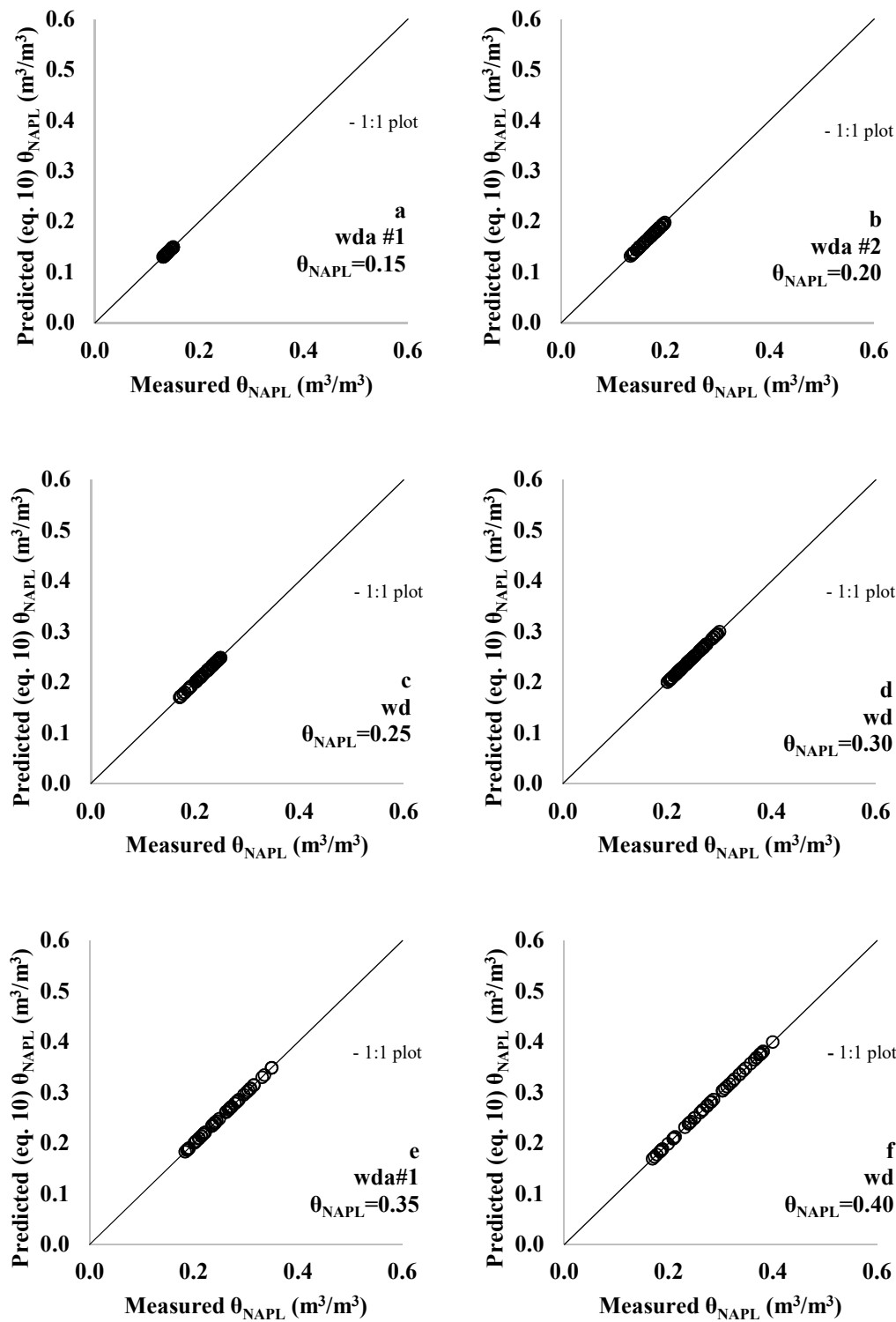

**Figure 5 a, b, c, d, e, f. Measured (equation 10) vs. known volumetric NAPL content ($\theta_{NAPL}$) of contaminated soils, with reference to the different remediation tests in figure 4.**

**Tables**

**Table 1**. Estimated $\alpha$ parameter of equation 10 for all three washing solutions (wd, wda#1 and wda#2) and volumetric NAPL content ($\theta_{NAPL}$) tested.

| parameter | washing solution | $\theta_{NAPL}$ | | | | | |
|---|---|---|---|---|---|---|---|
| | | 0.15 | 0.20 | 0.25 | 0.30 | 0.35 | 0.40 |
| $\alpha$ | wd | 0.45 | 0.30 | 0.49 | 0.65 | 0.67 | 0.55 |
| | wda#1 | 0.25 | 0.45 | 0.45 | 0.42 | 0.50 | 0.55 |
| | wda#2 | 0.20 | 0.45 | 0.30 | 0.45 | 0.55 | 0.52 |

**Table 2**. Model efficiency (*EF*) and mean bias error (*MBE*) statistical indices, referring to measured and predicted (equation 10) volumetric NAPL content ($\theta_{NAPL}$).

| Washing solution | $\theta_{NAPL}$=0.15 | | $\theta_{NAPL}$=0.20 | | $\theta_{NAPL}$=0.25 | |
|---|---|---|---|---|---|---|
| | EF | MBE | EF | MBE | EF | MBE |
| wd | 0.98 | 1.548 | 0.93 | -0.422 | 0.96 | 0.570 |
| wda#1 | 0.86 | 0.405 | 0.99 | 0.516 | 0.97 | -0.048 |
| wda#2 | 0.84 | 0.148 | 0.94 | 0.420 | 0.66 | 0001 |
| | | | | | | |
| Washing solution | $\theta_{NAPL}$=0.30 | | $\theta_{NAPL}$=0.35 | | $\theta_{NAPL}$=0.40 | |
| | EF | MBE | EF | MBE | EF | MBE |
| wd | 0.98 | -0.023 | 0.99 | -0.153 | 0.99 | -0.179 |
| wda#1 | 0.95 | -0.074 | 0.99 | -0.066 | 0.99 | 0.303 |
| wda#2 | 0.91 | 0.014 | 0.97 | 0.326 | 0.99 | 0.019 |

*Range of model applicability: 0.15$\leq \theta_{NAPL} \leq$0.40.