# Peer review of "A soil non-aqueous phase liquid (NAPL) flushing laboratory"

_Hydrology and Earth System Sciences, 2019_

## Referee Comment (RC1) · Ty P. A. Ferre (Referee) · 23 May 2019

I have to admit that I was fully expecting not to like this paper. The idea is very simple - using a dielectric mixing model to assess NAPL saturation in an otherwise water saturated medium. But, the simple approach worked, leaving me to wonder - why hasn't this been done before??

My only complaint is that the authors have not used original sources. There was a lot of work done on mixing models and rewriting these models as alpha mixing models

to determine fluid saturations. I would suggest that they avoid citing review articles (even though some of these will positively impact my H index) and look for the citations within these citations. Similarly, go to lengths to avoid self citation if you can find older references. I would recommend in particular that the authors look at references by Rosemary Knight, Dave Redman, and Timo Heimovaara.

The explanation of the rapid mobility is not as strong as it could be, although I don't know if this is critical. I suspect that the disagreement with the model has to do with the spatial, pore-scale distribution of NAPL during the initial flushing. This could point to difficulties in using this approach in other than homogeneous media, where NAPLS may collect along layer boundaries or in extended ganglia. But, again, this response is not central to the overall demonstrated ability to monitor NAPL concentrations with dielectrics.

In summary, I would like to congratulate the authors for filling a gap that I have not seen filled in this use of TDR.

Best Ty Ferre

---

## Referee Comment (RC2) · Magnus Persson (Referee) · 18 Jun 2019

General comments This paper deals with TDR measurements to asses different washing fluids to remove oil from soil samples. The topic should be of interest for the readers of the journal. In general the manuscript is well written, properly organized and in most parts easy to follow, although some awkard sentences could be found. However, the methodology needs to be better described (see comments below). I also miss a more thorough analysis and discussion of the results. A final language check would improve the paper even more. I have a few major comments listed here, and further some minor

comments listed below. My recommendation is that the paper can be published after a moderate revision.

Scientific questions 1. Title I would suggest removing "and modeling" from the title since the modeling part is not the main focus of the paper. Furthermore, no new model was developed, you only used an existing one.

2. Novelty/objectives Using TDR to measure NAPL content is not new, neither is the use of mixing models. Despite of this, very few applications outside controlled laboratory conditions have been published. Most published studies have merely showed the possibilities of using TDR without much practical focus of how this should be done and why. The topic itself is, thus, not very novel. The present paper, however, has its strengths in that TDR is used as a real time and in situ monitoring tool to asses different washing fluids. This is, to my knowledge, the first time this is done. This should be made more clear in the manuscript and this should be reflected in both the objectives and in the conclusion.

3. Using initially oven dried soil In (almost) every natural application, there would be both water and NAPL present in a contaminated soil. I can see why you chose to use a NAPL/dry soil mix to start with as this will lead to measurements that are relatively easy to interpret. But, you should at least discuss this in the manuscript and also point out what would need to be done differently is water also was present in the contaminated soil from the beginning.

4. Distribution of NAPL within the sampling volume The TDR reading is not only affected by the volumetric content of the components, but also the distribution of the components in the sampling volume. You discuss this in the section Model calibration and validation. However, I think you could elaborate this a bit. Especially with respect to my previous comment. How would water affect the distribution of the NAPLs in the contaminated soil and what does all this mean in terms of accuracy and applicability of the method? During the upward infiltration the NAPL distribution will change both

along and transverse the probe length. How is this affecting the measurements?

5. Saturated samples During the upward flow experiments the soil samples are saturated with NAPL and washing fluids. In any real application there will also be air present to some extent unless you are in the saturated zone. I think you should point this out.

technical corrections Line 14 and 15 Change "diverse" to "different" and "varying" in line 14 and 15, respectively Line 17 You use three different terms for the same thing; NAPL, hydrocarbon, and oil. I think you should stick to one term in order to avoid confusion. Line 107 Also include the value for water at 25 degrees. Line 116 What is the width of the TDR probe? Line 122 Include units for the NAPL content (m3/m3) to make clear that you are talking about volumetric content. Both here and in the entire manuscript. Line 129 How was the oil content of the effluent determined? Line 165 It is a little unclear which data you use for calculating the alpha parameters in Table 1. Is it the data from the leaching experiments? I guess it must be since no other experiment with both NAPL and washing fluid was described. But when you, e.g., say that the NAPL content is 0.4 m3/m3, this is only the NAPL content when the experiment starts. As the experiment goes on the NAPL content will decrease. When the NAPL content changes, also the alpha parameter is likely to change. Could you comment on this? And perhaps show the raw data and the model estimations to see the scatter in the data. Perhaps I misunderstand how you did this, if so please try to explain better. Line 169 This is the first time you mention "validation dataset". Which dataset is this? This implies that you also have a calibration dataset (which I guess you used to achieve the alpha parameters, see my comment above). Again, I think I do not fully understand which datasets you have. Line 163-187 This could be elaborated. What does the different alpha values mean, how do they correspond to other studies? Line 190-199 the conclusion could be elaborated a little. Table 1 the alpha value for wda#2 and NAPL content 0.20 (005) is very different from the others, can you explain why?

149, 2019.

---

## Referee Comment (RC3) · Anonymous Referee #3 · 24 Jun 2019

The manuscript presents a study where TDR methodology is used to measure NAPL concentration in soil during remediation treatments. The experimental setup is based TDR measurement of saturated column where the soil was initially mixed with variable amounts of NAPL (corn oil). The oil contamination ranged from 5 to 40% volumetrically, and the remediation was based on washing treatment with solutions containing water, detergent and methanol in three different ratios. Estimation of the oil content in the soil was done through standard permittivity measurement using TDR waveguides which provide the bulk dielectric permittivity of the soil. The measured results were compared

with the permittivity expected form dielectric mixing model which accounted for the specific contribution of the various materials (solid, water, detergent, methanol and NAPL). Obviously, the motivation for the study is directly related to the urgent need to develop measurements tolls for validation of remediation efficiency. Therefore, the objectives of this study are of very high environmental importance and relevancy to HESS readers. Nevertheless, some aspects in the manuscript require revision and an improved discussion. The motivation for this study, as presented across the manuscript, is related directly soil remediation from NAPLE. Yet the soil, which is the upper part of the subsurface, is characterized by everchanging water content. Since water content has the most significant impact on the bulk dielectric properties and the entire experimental setup refers saturated sediment, the use the term soil is misleading. Therefore, the presented method should be limited to saturated porous medium/sediment (aquifer) and not to soil. It is not clear whether the manuscript focuses on using TDR to measure the NAPL content in the sediment or on the efficiency of the treatment method. If the author wishes to test the TDR efficiency to measure NAPL concentration, then the experiment provided only qualitative data showing the reduction in NAPL during the early stages of the washing phase. Yet it has been shown only in the vary high oil concentration >15% volumetric content (figure 3). Although the experiment included lower concentration range of 5 and 10 % results from these tests were not presented or discussed although from environmental point of view these are very high concentrations. The authors show that the TDR results are biased, compare to the model (figure 4) and suggested that the reason is related to the flow and transport mechanism within the sediment column, where trapped oil turns immobile and therefor unwashable. Trapped oil in porous domain is a known phenomenon. Nevertheless, it is not clear why the TDR, which measures the bulk dielectric properties of the domain, is affected from the flow and transport mechanism. It should see total weighted contribution of all component where it is trapped or mobile. The authors choose to demonstrate the washing effect of different solution on NAPL removal from the sediment using the TDR. However, using 60 pore volumes to wash the soil is totally non reasonable or

realistic by any means. In other words, if the authors wish to establish the TDR method as a tool for measuring NAPL content in the sediment they have to separate the washing effect from the concentration measurement. As such the NAPL concentration and the model calibration will be unbiased and more efficient. For example, biodegradation method would work much faster and provide quicker results.

---

## Author Comment (AC1) · 2 Jul 2019

Dear Prof. Ferré,

With reference to the paper: hess 2019-149, by A. Comegna et al., please find below the replies to your review. The authors would like to thank prof. Ferré for the invaluable review.

In accordance with your comments, we have rearranged the references in order to include some other relevant manuscripts that were, not intentionally, overlooked. Fur-

thermore in the Model calibration and validation section, following your suggestion, we commented on the problem related to the initial pore-scale distribution of NAPL in the soil sample, which could play a role, with the "rapid mobility of the fluid" at the beginning of the removal experiment. PS: following the Journal submission procedure, the revised version of our paper will be uploaded after the interactive discussion session has been closed.

Sincerely The authors

---

## Author Response (AR1)

**Dear prof. Vanclooster,**

please find below the replies to the referees, and the marked-up version of our manuscript.

Thank you

**Dear Prof. Ferré,**

With reference to the paper: **hess 2019-149**, by A. Comegna et al., please find below the replies to your review. The authors would like to thank prof. Ferré for the invaluable review.

In accordance with your comments, we have rearranged the references in order to include some other relevant manuscripts that were, not intentionally, overlooked. Furthermore in the **Model calibration**

**and validation** section, following your suggestion, we commented on the problem related to the initial pore-scale distribution of NAPL in the soil sample, which could play a role, with the "*rapid*

*mobility of the fluid*" at the beginning of the removal experiment.

**Dear Prof. Persson,**

With reference to the paper: **hess 2019-149**, by A. Comegna et al., please find below the replies to your review.

The authors would like to thank Prof. Persson for his useful suggestions which have been **fully**

**accepted**. We explain below how the revised paper was reorganized.

**- Major Comments:**

1. …**and modeling** was removed from the title. Moreover, in order to eliminate any misunderstandings about the development of a new dielectric mixing model (we have only rewritten it for our purposes), we substituted in the text (line 18) the term *develop* with

*calibrate and validate*, and (line 66) the term *build* with *revisit*.

2. In accordance with your comment we emphasized in the text (**introduction section**: lines 45-

68) the novelty of the present research.

3. You are right with reference to the possibility of investigating, during remediation, the dielectric response of an initially four-phase medium (i.e. soil+NAPL+water+air), but (as you already wrote in your review) the present research is a first attempt to monitor in real time (with TDR) the dielectric response of the medium during a decontamination process. Thus we chose a simple initial scenario to avoid possible dielectric "interferences" related with other phases. This aspect could be explored in further research (a specific sentence regarding this possibility was introduced in the conclusion). Anyway to carry out our research we followed the approach of **Francisca and Rinaldi (2006)**, who published a paper entitled*: Removal of*

*immiscible contaminants from sandy soils monitored by means of dielectric measurements*

(doi: 10.106/(ASCE)0733-9372(2006)132:8(931)).

4. I agree with you that the dielectric response of a multiphase medium depends not only on the

NAPL (and eventually water) volumetric content, but is also influenced by their internal distribution; **In accordance with your comment we sought to emphasize this aspect in the**

**text** (lines 206-209) . Moreover I would like to stress the fact that TDR (as you already know)

cannot allow us to infer how fluid distribution affects dielectric measurement; this aspect could be a further research topic, which should be developed by coupling TDR with different geophysical methods, such as the *Gamma Ray Attenuation* technique, that gives more accurate information on fluid distribution within the contaminated soil sample.

**- Technical corrections:**

1. In accordance with your comment we changed *diverse* to *different* (line 14) and *diverse* to

*varying* (line 15).

2. In accordance with your comment we substituted in the abstract (and where possible in the whole manuscript) the terms *hydrocarbon* and *oil* with NAPL.

3. In accordance with your comment we introduce in the paper the dimensions of the TDR probe (line 136).

4. In accordance with your comment we introduced in the text the dimension of the volumetric

NAPL content $\theta_{NAPL}$: $m^3/m^3$ (line 19).

5. In accordance with your comment we better commented in the paper how was the oil content determined on the effluent (lines 148-150) and in agreement with the actual description, we modified figure 1.

6. In accordance with your comment we better describe how the $\alpha$ parameters were determined (lines 187-188). Furthermore, we made some new comments in the paper regarding the

*calibration and validation* data set that we employed for model calibration and validation (lines 151-152). Finally, we introduce in the text (line 143) the term **initial**, in order to specify that: $\theta_{NAPL}$=0.15, 0.20, 0.25, 0.30, 0.35 and 0.40 was the volumetric NAPL content at the beginning of the different experiments conducted.

7. With reference to parameter $\alpha$ in the dielectric mixing model adopted, I would like to emphasize that $\alpha$, in our application is a pure fitting parameter, obtained from the calibration data set. This means that for a fixed $\theta_{NAPL}$ value and washing solution, the dielectric model was fitted (using a least square algorithm) to the whole set of experimental calibration data (i.e. the data obtained from the beginning to the end of the remediation test). For this reason

$\alpha$ must be considered constant. This aspect is now commented in the **Model calibration and**

**validation** section (lines 187-188).

8. In accordance with your comment we introduce a series of 1:1 scatter plots (figure 5a, b, c, d,e, f).

9. See comment #6.

10. In accordance with your comment we elaborated the section **Model calibration and**

**validation**.

11. No more comments can be made in the manuscript with reference to parameter $\alpha$ for the reasons of comment #6.

12. In accordance with your comment we revisited the conclusions.

13. In Table 1 the $\alpha=0.05$ value for wda#2 and $\theta_{NAPL}=0.20$ was wrong. Thank you for your observation. The correct ($\alpha=0.45$) value was inserted.

**Dear Referee #3,**

With reference to the paper: **hess 2019-149**, by A. Comegna et al., please find below the replies to your review. The authors would like to thank you for your comments on our paper. We would like to say that part of your observations have been resolved in the revised version of the manuscript, in accordance with those of the other two referees.

1. That said, we would like to stress the fact that the purpose of this study was to investigate a possible extension of TDR technology to assess the effects of NAPL removal in soil organic mixtures, "*in real time*" during a decontamination process. As is well known, TDR is one of the most important geophysical methods, with its first applications in Soil Physics in 1980

(see Topp et al., 1980). In recent years several efforts have been made to extend the use of

TDR technology. See for example Kachanoski et al. (1992) who employed TDR for measuring in the "soil" the presence of a leaching solute. With direct reference to TDR-NAPL

applications, most studies have demonstrated the potential of the TDR technique in estimating

NAPL presence in saturated soils (Redman and DeRyck, 1994; Chenaf and Amara, 2001;

Haridy et al., 2004; Mohamed and Said, 2005; Moroizumi and Sasaki, 2008). Some experiments have been conducted on unsaturated soils (Persson and Berndtsson, 2002; Rinaldi and Francisca, 2006; Francisca and Montoro, 2012). In these studies, the estimation of NAPLs using TDR measurements of dielectric properties has relied greatly on various mixing models relating the measured dielectric permittivity to the volume fractions of the pore fluids and various soil phases such as solid, water, air, and NAPLs (van Dam et al., 2005). Finally, I

would like to recall the papers of Comegna et al. (2016) and Comegna et al. (2017) which tackled the problem of NAPL detection in variously saturated homogeneous and layered

"soils", respectively.

2. As already stated in the manuscript, the range of model applicability is: $0.15<\theta_{NAPL}<0.40$. At

$\theta_{NAPL}=0.05$ and 0.10, TDR is not sensitive to NAPL volumes.

3.   The present research is a first attempt to monitor via TDR the dielectric response of an NAPL-

 contaminated medium during a decontamination process. Thus we chose a simple initial

 scenario to avoid possible dielectric "interferences" related to other phases. This aspect could

 be explored in further research.

                                    Sincerely

                                The authors

[revised manuscript text omitted]

a) θ_NAPL=0.15

b) θ_NAPL=0.20

c) $\theta_{NAPL}=0.25$

[Figure]

d) $\theta_{NAPL}=0.30$

[Figure]

[Figure]

**Figure 2. Selection of experimental relationships between the measured dielectric permittivity**

**($\varepsilon_{s\text{-}ws\text{-}NAPL}$) and number of pore volumes $T$ under the effect of different washing solutions: i)**

**water-detergent (wd) and ii) water-detergent-alcohol (wda#1 and wda#2).**

[Figure]

Figure 3. Volume of NAPL recovered ($V_{NAPL\text{-}Rem}$) with respect to the initial volume of NAPL

present in the soil sample ($V_0$) of different washing solutions (wd, wda#1 and wda#2) for different experiments ($\theta_{NAPL}$=0.15, 0.20, 0.25, 0.30, 0.35, 0.40).

[Figure]

[Figure]

[Figure]

[Figure]

[Figure]

**Figure 4 a, b, c, d, e, f. Selection of observed (symbols) and modeled (dashed lines) volumetric**

**NAPL content ($\theta_{NAPL}$) versus dielectric permittivity ($\varepsilon_{s\text{-}ws\text{-}NAPL}$), with reference to the three**

**washing solutions (wd, wda#1 and wda#2) used during the remediation tests.**

[Figure]

Figure 5 a, b, c, d, e, f. Measured (equation 10) vs. known volumetric NAPL content (θNAPL) of contaminated soils, with reference to the different remediation tests of figures 4.

**Tables**

**Table 1**. Estimated $\alpha$ parameter of equation 10 for all three washing solutions (wd, wda#1 and wda#2) and volumetric NAPL content ($\theta_{NAPL}$) tested.

| parameter | washing solution | $\theta_{NAPL}$ | | | | | |
|-----------|------------------|------|------|------|------|------|------|
| | | **0.15** | **0.20** | **0.25** | **0.30** | **0.35** | **0.40** |
| $\alpha$ | wd | 0.45 | 0.30 | 0.49 | 0.65 | 0.67 | 0.55 |
| | wda#1 | 0.25 | 0.45 | 0.45 | 0.42 | 0.50 | 0.55 |
| | wda#2 | 0.20 | 0.45 | 0.30 | 0.45 | 0.55 | 0.52 |

**Table 2**. Model efficiency (*EF*) and mean bias error (*MBE*) statistical indices, referring to measured and predicted (equation 10) volumetric NAPL content ($\theta_{NAPL}$).

| Washing solution | $\theta_{NAPL}$=0.15 | | $\theta_{NAPL}$=0.20 | | $\theta_{NAPL}$=0.25 | |
|------------------|------|------|------|------|------|------|
| | **EF** | **MBE** | **EF** | **MBE** | **EF** | **MBE** |
| wd | 0.98 | 1.548 | 0.93 | -0.422 | 0.96 | 0.570 |
| wda#1 | 0.86 | 0.405 | 0.99 | 0.516 | 0.97 | -0.048 |
| wda#2 | 0.84 | 0.148 | 0.94 | 0.420 | 0.66 | 0001 |
| | | | | | | |
| **Washing solution** | **$\theta_{NAPL}$=0.30** | | **$\theta_{NAPL}$=0.35** | | **$\theta_{NAPL}$=0.40** | |
| | **EF** | **MBE** | **EF** | **MBE** | **EF** | **MBE** |
| wd | 0.98 | -0.023 | 0.99 | -0.153 | 0.99 | -0.179 |
| wda#1 | 0.95 | -0.074 | 0.99 | -0.066 | 0.99 | 0.303 |
| wda#2 | 0.91 | 0.014 | 0.97 | 0.326 | 0.99 | 0.019 |

*Range of model applicability: $0.15 \leq \theta_{NAPL} \leq 0.40$.